# Risk Factors, Pathogenesis, and Strategies for Hepatocellular Carcinoma Prevention: Emphasis on Secondary Prevention and Its Translational Challenges

**DOI:** 10.3390/jcm9123817

**Published:** 2020-11-25

**Authors:** Shen Li, Antonio Saviano, Derek J. Erstad, Yujin Hoshida, Bryan C. Fuchs, Thomas Baumert, Kenneth K. Tanabe

**Affiliations:** 1Division of Surgical Oncology, Massachusetts General Hospital Cancer Center and Harvard Medical School, Boston, MA 02114, USA; sli8@bidmc.harvard.edu (S.L.); DERSTAD@partners.org (D.J.E.); Bryanfuchs1@gmail.com (B.C.F.); 2Inserm, U1110, Institut de Recherche sur les Maladies Virales et Hépatiques, Université de Strasbourg, 67000 Strasbourg, France; saviano@unistra.fr; 3Simmons Comprehensive Cancer Center, UT Southwestern Medical Center, Department of Internal Medicine, Dallas, TX 75390, USA; Yujin.Hoshida@utsouthwestern.edu

**Keywords:** hepatocellular carcinoma, hepatitis, NASH, chemoprevention

## Abstract

Hepatocellular carcinoma (HCC) is a leading cause of cancer-associated mortality globally. Given the limited therapeutic efficacy in advanced HCC, prevention of HCC carcinogenesis could serve as an effective strategy. Patients with chronic fibrosis due to viral or metabolic etiologies are at a high risk of developing HCC. Primary prevention seeks to eliminate cancer predisposing risk factors while tertiary prevention aims to prevent HCC recurrence. Secondary prevention targets patients with baseline chronic liver disease. Various epidemiological and experimental studies have identified candidates for secondary prevention—both etiology-specific and generic prevention strategies—including statins, aspirin, and anti-diabetic drugs. The introduction of multi-cell based omics analysis along with better characterization of the hepatic microenvironment will further facilitate the identification of targets for prevention. In this review, we will summarize HCC risk factors, pathogenesis, and discuss strategies of HCC prevention. We will focus on secondary prevention and also discuss current challenges in translating experimental work into clinical practice.

## 1. Introduction

Hepatocellular carcinoma (HCC) is the fifth most common malignancy and the fourth leading cause of cancer-associated mortality worldwide [1,2]. In 2015, there were 854,000 cases of HCC globally, with a male-to-female ratio of 2.4:1 [3]. Sub-Saharan Africa and East Asia have the highest incidence rate of HCC (more than 20 per 100,000 individuals), while North America and Europe have lower incidence levels (less than 5 per 100,000 individuals) [4,5,6]. Between 1996 and 2006, the HCC incidence rate in the US surveillance, epidemiology and end results (SEER) registries increased from 3.1 to 5.1 per 100,000 individuals, while the US liver cancer mortality rose from 3.3 to 4.0 per 100,000 individuals [7]. From 2006 to 2017, the rate of HCC has increased by 2–3% annually, largely due to the high prevalence of hepatitis C virus (HCV) cirrhosis and nonalcoholic steatohepatitis (NASH) [8,9].

Early HCC diagnosis occurs in 30–60% of cases, enabling curative treatments such as surgical resection and liver transplantation. Even with curative approaches, HCC recurrence is observed in up to 80% of patients within 5 years [10]. In advanced disease, surgical and systemic therapies have largely failed to yield survival benefits [11]. Until recently, Sorafenib was the only FDA-approved agent for advanced HCC. Since 2017, other multi-kinase inhibitors have been approved for second-line treatments, such as Cabozantinib and Ramucirumab [12,13,14]. Checkpoint inhibitors such as Nivolumab [15] and Pembrolizumab [16] have been either FDA approved or under investigation. Importantly, systemic therapies have substantial adverse effects that are difficult to manage in cirrhotic patients [17]. Finally, the high costs of approved therapies limit their use in low-resource countries [18]. Given the potential to identify high risk individuals and low survival rate once diagnosed, HCC prevention in at-risk patients can be a successful and alternative approach for HCC management. Here, we will review HCC risk factors, pathogenesis and current strategies for prevention, specifically secondary prevention and its clinical challenges.

## 2. Risk Factors and Carcinogenesis

### 2.1. Hepatitis B Virus (HBV)

Globally, 2 billion people have been exposed to HBV along with 250–350 million chronic carriers [19]. In high endemic areas, the HBV carrier rate is nearly 8% [20]. In areas of high incidence, 80% of patients with HCC are seropositive for the hepatitis B surface antigen (HBsAg) [21]. In East Asia, the HCC incidence rate in chronic HBV carriers ranges from 0.6 per 100 individuals without cirrhosis to 3.7 per 100 individuals with compensated cirrhosis. In Europe and the United states, the incidence rate ranges from 0.3 per 100 individuals without cirrhosis to 2.2 per 100 individuals in subjects with cirrhosis [22,23]. In addition, 10–20% of patients with HBV can develop HCC in the absence of cirrhosis [24]. Other factors that have been reported to increase the risk of HCC include duration of infection, viral load, environmental exposures (aflatoxin, alcohol, or tobacco), demographic features such as male sex, older age, family history of HCC, and co-infections with HCV, HDV, and HIV [25,26].

Two major mechanisms of HBV-induced HCC carcinogenesis have been proposed. First, chronic HBV can induce cirrhosis through the activation of HBV-specific T-cells, chemokine-mediated neutrophils, macrophages, and natural-killers [27,28]. These inflammatory cells promote carcinogenesis by stimulating hepatocyte regeneration, reactive oxygen species (ROSs) production, and DNA damage. Second, HBV DNA can be integrated into the host genome, prompting insertional activation of proto-oncogenes [29], induction of chromosomal instability [30], and the transcription of pro-carcinogenic HBV genes such as truncated envelope proteins [31], hepatitis B X gene (HBx) [32], and hepatitis B spliced proteins [33].

### 2.2. Hepatitis C Virus (HCV)

HCC induced by HCV has the highest HCC mortality rates per 100,000 individuals in the United States [34,35]. Chronic HCV is most prevalent in the “baby boomer” generation, defined as adults born between 1945 and 1965 who were exposed to blood transfusions, clotting factors, and hemodialysis prior to 1992 [36]. Moreover, the current opioid epidemic further contributes to the spread of HCV [37]. Between 2010 and 2017, a three-fold increase in acute HCV infections was reported by the CDC resulting from the opioid epidemic [38]. The annual incidence of HCV related HCC in patients with cirrhosis is extremely high, ranging from 1 to 12% per year [39,40]. In a large cohort of US patients with HCV, patients with genotype 3 are more likely to develop cirrhosis and HCC than other HCV genotypes [41].

HCC development in HCV is primarily associated with fibrosis and the viral copy number [42]. HCV associated HCC development occurs in a stepwise fashion, typically spanning over decades. All cases of HCV HCC arise from mutations in hepatocytes within a cirrhotic background. HCV proteins have also been shown to promote cellular proliferation, transformation, and tumor growth. Over-expression of HCV core proteins, NS3 and NS5A, inhibit tumor suppressor genes *TP53*, *TP73*, and *RB1*, as well as negative cell cycle regulators such as CDKN1A. E2 and NS5B activate the RAF/mitogen-activated protein kinase (MAPK)/ERK kinase pathways [43]. NS5A activates the PI3K/AKT and beta-catenin/WNT pathways, and evades apoptosis by caspase-3 inhibition [44]. It is important to note that the interaction between nonstructural protein NS5A and HCV is dependent on Rab18-positive lipid droplets [45].

### 2.3. Nonalcoholic Fatty Liver Disease (NAFLD)/Nonalcoholic Steatohepatitis (NASH)

NAFLD/NASH has emerged as a leading cause of end-stage liver disease as well as HCC. Studies have demonstrated that the incidence of HCC in patients with NASH ranges from 2.4% over 7 years to 12.8% over 3 years [46]. Recent studies in the US have shown that NAFLD/NASH-related mortality has dramatically increased in the last 10 years together with NAFLD/NASH-related liver cirrhosis [47]. HCC in NAFLD/NASH is often diagnosed in patients without cirrhosis and is associated with late onset diagnosis and a higher tumor burden [48]. Moreover, patients with NASH receive sub-optimal HCC surveillance in comparison to patients with HCV cirrhosis [49].

The mechanism of HCC carcinogenesis as a result of NAFLD is not completely clear. Steatosis alone is not a driver of HCC as chronic inflammation is necessary for carcinogenesis [50]. Fat- tissue-derived free fatty acids (FFAs) lead to steatosis and lobular inflammation through the activation of intrahepatic lymphocytes and infiltrating macrophages. Hepatocyte cell death and compensatory proliferation together with increasing levels of tissue necrosis factor (TNF) superfamily members, transforming growth factor β (TGF-β), activation of hepatic stellate/liver sinusoidal endothelial cells, and hepatocyte chromosomal aberrations all contribute to HCC development [51]. The increased hepatocyte metabolism and oxidation of fatty acid induce overproduction of ROSs [52]. The excess of triglycerides and FFAs impair the initiation of autophagy through the activation of mammalian target of rapamycin (mTOR). When the antioxidant capacity of the hepatocytes is exceeded, DNA damage and oxidation occurs, eventually resulting in cell death [53,54].

It has also been suggested that the inflammatory responses seen in patients with NASH might be caused by an increase in gut permeability. Even though it is unclear whether a leaky gut is the consequence or the cause of NASH, it is evident that the translocation of lipopolysaccharide from gram-negative bacteria is an important aggravating factor for liver inflammation and fibrosis [55].

### 2.4. Lifestyle Risk Factors

Alcoholic liver disease (ALD) alters hepatic metabolism, causing progressive steatosis, fibrosis/cirrhosis, and HCC [56,57]. The risk of HCC increases with alcohol consumption as low as 10 g/day [58]. Perrsen et al. found that consuming more than three drinks daily was associated with an increase in HCC incidence and liver disease-related mortality [59]. There is a synergistic relationship between alcohol use of >60 g per day and viral hepatitis, with an approximately two-fold increase in the odds-ratio of developing HCC [60]. Alcohol use at least four times per week annually along with obesity (BMI > 30) increases the incidence of HCC [61]. Kimura et al. found that mild alcohol use (<20 g/day) in patients with NASH and advanced fibrosis was associated with a significant increase in the risk of HCC [62], while Ochiai et al. showed that ethanol intake ≥ 40 g was associated with a significant increase in multinodular HCCs [63].

The mechanism of ALD induced HCC is partially understood. Alcohol consumption can alter metabolic pathways including fatty acid oxidation and lipogenesis. Chronic alcohol consumption leads to an abnormal accumulation of acetaldehyde, which can exert carcinogenic effects through the formation of DNA-protein adduct [64]. Acetaldehyde has been shown in vitro to interfere with the transcriptional activities of peroxisome proliferator activated receptors (PPARs) and sterol regulatory element binding protein 1 (SREBP-1) [65,66]. Alcohol consumption can also reduce the level of 5′ AMP-activated protein kinase (AMPK), an important regulator of lipogenesis [67]. It is important to note that the severity of ALD is associated with genetic susceptibility. Genome-wide association studies (GWASs) have identified genetic risk loci for ALD, including *PNPLA3* [68] and *MBOAT7/TMC4* being related to a higher risk of cirrhosis in alcohol abusers [69].

Smoking is another important lifestyle risk factor for HCC. Tobacco smoking contributes to 13% of all HCC cases globally [70]. Current smokers have higher risks of HCC (hazard ratio 1.86, 95% CI: 1.57–2.20) [71], while those who quit for over 30 years have similar risks to non-smokers [72]. Tobacco contains multiple carcinogenic agents, including aromatic hydrocarbons [73], diethylnitrosamine (DEN) [74], and 4-aminobiphenyl [75]. Tobacco use is associated with an increase in inflammatory cytokines and ROS [76]. Tobacco also has been shown in rodent models to exacerbate the severity of NAFLD through the increasing of oxidative stress and hepatocellular apoptosis [77].

### 2.5. Environmental Carcinogens

A number of environmental chemicals have been implicated in HCC carcinogenesis. The best documented are aflatoxins. Other factors include vinyl chloride, arsenic compounds, polychlorinated biphenyls, and radioactive compounds [78]. Aflatoxins, mycotoxins produced by *Aspergillus flavus* and *Aspergillus parasiticus*, are frequently found in contaminated grain products such as maize and ground nuts in farming communities in sub-Saharan Africa, South America, and parts of Eastern Asia [79,80]. Aflatoxin B1 has been shown to form DNA adducts with hepatic DNA, leading to carcinogenesis in both humans and animal models [81]. In regions with high aflatoxin exposure, a 70-fold increase in the risk of HCC development has been observed [82].

### 2.6. Genetic Predisposition

Alpha 1-antitrypsin (AAT) deficiency is an autosomal recessive disease that results from mutations in the *SERPINA1* gene. This gene encodes a serine protease inhibitor, which functions to inhibit neutrophil elastase. A retrospective study in Sweden found an odds ratio of 20 for the development of HCC in patients with AAT deficiency [83]. Glycogen storage disease I, or Von Gierke’s disease, leads to the impairment of glucose-6-phosphatase activity with excess glycogen storage in the liver [84]. Patients with glycogen storage disease I can develop hepatocellular adenomas by their second or third decade of life. A number of these patients go on to develop HCC [85]. The risk of HCC in patients with hemochromatosis is approximately 20 times higher than the general population [86]. Lastly, hereditary tyrosinemia type I is an autosomal recessive disease caused by an enzymatic deficiency in the catabolic pathway of tyrosine [87]. This disease can lead to acute hepatic failure or cirrhosis in infancy. In addition, 40% of patients who survive beyond the age of 2 develop HCCs [88].

GWAS have identified single-nucleotide polymorphisms (SNPs) that are associated with HCC carcinogenesis. A SNP in the epidermal growth factor (*EGF*) gene (rs4444903) was associated with an elevated risk of HCC in patients with cirrhosis [89]. A SNP (rs17401966) in Kinesin family member 1B (*KIF1B*) was associated with HBV-related HCC. Other SNPs in the Ubiquitination factor E4B (*UBE4B*) and Phosphogluconate dehydrogenase (*PDG*) genes were also shown to be associated with HCC amongst HBV positive patients [90]. Two SNPs (rs2596542 and rs1012068) discovered in a GWAS conducted in two large Japanese cohorts were significantly related with HCV-induced HCC [91,92].

### 2.7. Endocrine Risk Factors

Thyroid hormones are essential for lipid metabolism and have been shown to play a role in the pathogenesis of NAFLD/NASH [93]. Hypothyroidism has been demonstrated to be more common in patients with HCV, with a higher prevalence in those with cirrhosis [94]. Studies have also shown that patients with hypothyroidism have a two-fold higher risk of HCC than those with no prior history of thyroid cancer [95,96]. High thyroid stimulating hormone levels in HCC patients were found to be associated with larger tumor sizes [97]. Huang et al. demonstrated that 3,3′5-tri-iodo-l-thyronine (T_3_) suppressed HCC cell proliferation through the inhibition of serine/threonine-protein kinase, PIM-1, via miRNA (miR-214-3p) [98]. T3 supplementation in rats resulted in fewer tumor nodules as well as a shift in global transcriptomic expression profile. T3 was shown to exert anti-carcinogenic effects through the maintaining of genes responsible for hepatocyte differentiation, such as *KLF9* and *HNF4a* [99].

Epidemiologically, HCC predominately occurs in males, with a male-to-female ratio ranging from 1.5:1 to 11:1. HCC prognosis, survival, and disease free survival after surgery are significantly better in females than males [100]. The predominance of HCC in males has been thought to be related to the effects of androgen/androgen receptors (ARs). ARs have been shown to promote HBV viral replication and HBV induced HCC [101], while AR knockout mice have fewer tumor nodules [102]. AR signaling has been demonstrated to promote key regulators of HCC carcinogenesis, including the MAPK/STAT/AKT pathway [103].

Estrogen/estrogen receptors (ERs) have been found to have protective effects against HCC while postmenopausal females have higher incidences of HCC [104,105]. Estrogen administration has been shown to reduce proinflammatory cytokines such as IL-6, a critical cytokine in HCC carcinogenesis [106]. Naugler et al. reported that estrogen treatment could reduce HCC carcinogenesis in DEN-injured rats by attenuating MyD88-dependent NF- κB signaling and inhibiting IL-6 signaling [107]. ER activation has also been shown to reduce STAT3 activation [108], a key regulator of the inflammatory tumor microenvironment [109].

## 3. Pathogenesis of HCC

A large body of research has been performed to address HCC pathogenesis. Large-scale genomic quantitative comparisons of HCC tumors have revealed the occurrence of chromosomal and microsatellite instability [110]. Loss of heterozygosity and SNP arrays have shown loss or mutations in tumor suppressor genes such as TP53 (*P53*) [111], retinoblastoma RB1 (*RB1*) [112], *CDKN2A* (*P16^INK4A^*) [113], and insulin-like growth factor-2 receptor (*IGF-2R*) [114]. Gain of function mutations such as *CTNNBI* (β-catenin) can upregulate the transcription of MYC, cyclin D1, and COX2 [115]. There is a strong association between HBV encoded viral protein HBx and the suppression of *P53* induced apoptosis [116]. HCV core protein can also have direct carcinogenic effects by inducing ROSs [117].

Dysregulations of miRNAs, a class of small non-coding RNAs, can lead to HCC carcinogenesis [118]. Gene expression profiling has revealed that miR-181 upregulation is associated with the Wnt/B-catenin pathway [119]. MiR-26 downregulation has been shown to be associated with poor prognosis and a higher risk for metastasis [120]. Silencing of miR-122 was associated with increased cancer invasion, elevated alpha-fetoprotein expression, as well as higher HCC grades [121].

Genome-wide gene expression profiling has been used to capture dysregulated gene-expression signatures [122]. Numerous genome-wide expression studies have identified molecular sub-classes of HCC [123,124]. Aggressive HCC tumors are characterized by increased genetic instability, cellular proliferation, and impairment of tumor suppressor genes [125]. Hoshida et al. categorized HCC into three classes. S1 tumors are the most aggressive and are characterized by higher activation of *TGF*-β. S2 tumors overexpress EPCAM, AFP, and IGF-2 [126], while S3 tumors have matured hepatocyte-like phenotypes [127]. Zucman-Rossi et al. characterized proliferative vs. non-proliferative sub-classes of HCC. The main traits of the proliferative subclass are related to tumor proliferation and survival, while non-proliferative HCCs resemble normal hepatocytes [128].

It is also important to recognize that the hepatic microenvironment significantly promotes tumor progression [129] and concomitantly limits therapeutic interventions [130]. The normal liver stroma maintains tissue integrity and acts as a barrier against tumor formation [131]. During chronic inflammation, a modified stroma is formed, enriched in carcinoma-associated fibroblasts [132,133] and tumor-associated immune cells [134,135]. In such a pro-carcinogenic environment, cancer cells are potentiated to grow and proliferate such that responses to conventional treatments are altered (Figure 1).

## 4. Molecular Biomarkers of HCC—The Prognostic Liver Signature

The major challenge in managing HCC is the complex and elusive mechanism of HCC carcinogenesis, leading to a scarcity of cancer biomarkers for targeted prevention trials. To circumvent this obstacle, a reverse engineering approach was developed to identify carcinogenic targets using long-term clinical follow-up patient cohorts, subsequently verified using in silico, in vitro, and in vivo models (Figure 2). It is hypothesized that cirrhosis leads to field cancerization, whereby cirrhotic liver tissue can harbor gene-expression signatures associated with carcinogenesis or recurrence after resection [136].

To verify this hypothesis, Hoshida et al. analyzed liver tissues surrounding resected HCV HCC tumors in 106 formalin-fixed, paraffin embedded blocks and identified a prognostic liver signature (PLS) containing 186 genes [137]. The poor-prognosis signature was found to be associated with an increase in liver-related deaths, progression of the Child–Pugh class, as well as HCC development. The 10-year HCC development rates were 42% and 18% for patients with poor and good prognostic signatures, respectively [138]. Though initially verified in HCV patients, the PLS also demonstrated significant concordance in liver tissues from HBV, alcohol, and NAFLD/NASH patients followed for 23 years [139]. This prognostic signature successfully verified the chemopreventive effect of erlotinib, a small molecule EGF pathway inhibitor, in multiple rodent models [140], and also led to the initiation of a cancer chemoprevention clinical trial (NCT02273362).

Besides the EGF pathway, other inflammatory and fibrotic pathways have been identified as valuable cancer targets for prevention. Top enriched regulator genes were *AKT1*, *SLC35A1*, *DDX42*, *ILK*, and *LPAR1*. AKT-activated mTOR inhibitors, including everolimus and sirolimus, are currently being investigated for chemoprevention after transplantation [141]. Lysophosphatidic acid receptor 1 (LPAR1) is the receptor for the bioactive lipid lysophosphatidic acid (LPA) produced from lysophosphatidyl choline (LPC) through the actions of a secrete lysophospholipase D named autotaxin (ATX). LPAR1 overexpression has been shown to promote fibrosis [142], inflammation and HCC carcinogenesis via upregulation of its downstream effectors, including RhoA/ROCK, RAS/MAPK/ERK, and AKT/PI3K (Figure 3) [143]. In a DEN model of cirrhosis, LPAR1 upregulation coincided with the development of cirrhosis. Furthermore, LPAR1 inhibition with ATX inhibitors attenuated liver fibrosis, reduced the number of HCC nodules, and reversed the PLS risk gene signature [137,144].

The reverse-engineering technique along with the transcriptomic analysis of cancer-prone markers can be used to not only unearth key biomarkers of cancer prevention, but also be used for the proof-of-concept of other experimental compounds. Inhibition of chromatin reader Bromodomain 4—a target identified by reverse-engineering—by use of a small molecule, JQ1, reduced HCC carcinogenesis in experimental rodent models by reverting the epigenetic as well as the poor prognostic signature [145]. Villa et al. found a five-gene signature that predicted tumor doubling time as well as overall survival. In this study, ultrasound surveillance was used to identify newly diagnosed HCCs in cirrhotic patients. Patients then underwent two CT scans 6 weeks apart in order to determine tumor doubling time. In this study, five genes (*ANGPT2*, *NETO2*, *ESM1*, *NR4A1*, and *DLL4*) that regulated angiogenesis and endothelial cell migration were significantly upregulated and predicted tumor doubling time and survival [146].

## 5. Prevention Strategies

Understanding the risk factors and pathogenesis of HCC provides an opportunity for prevention strategies. Prevention can be sub-divided into primary, secondary, and tertiary prevention. Primary prevention focuses on eliminating cancer-predisposing factors through early vaccination, lifestyle modifications, as well as environmental interventions. Globally, a significant reduction in the incidence of HCC was observed after the implementation of hepatitis B vaccination [147,148,149]. Preventive actions against HCV can be taken through changes in social/cultural/medical practices such as the prevention of IV drug use and efficient screening of blood products and medical instruments [150]. In Australia, a substantial decline in the estimated intravenous drug use resulted in a decline in the number of new HCV infections from 14,000 per year in 2000 to 10,000 per year in 2005 [151]. Regulations of environmental carcinogens, such as Aflatoxin through information dissemination, have significantly reduced the Aflatoxin level in endemic areas [152].

Secondary prevention aims to delay the progression of chronic liver disease. This approach strives to eradicate the etiological agents (HBV and HCV) or inhibit the various steps in the carcinogenic progression. In general, chemoprevention agents should be inexpensive, well tolerated for long-term treatment, and available to the general population.

Tertiary prevention targets cancer recurrence or de-novo carcinogenesis within 1–2 years after curative treatment [153]. Ikeda et al. demonstrated in a randomized control trial that interferon-based immunotherapy after HCC resection resulted in a significant reduction in HCC recurrence [154]. Mazzaferro et al. demonstrated that adjuvant interferon therapy may reduce late recurrence of HCC [155]. Post-operative interferon-alpha is currently being investigated in patients with low miR-26 expression after HCC resection (NCT01681446). Immunosuppression with mTOR inhibitors has also been shown to reduce HCC recurrence [156,157] (Figure 4).

## 6. Early Diagnosis and Surveillance

Compliance to HCC surveillance is associated with early diagnosis, allocation of curative treatment, and longer adjusted overall survival [158]. Practice guidelines from The American Association for the Study of Liver Diseases (AASLD) and The European Association for the Study of the Liver (EASL) recommend HCC surveillance for high risk patients by abdominal ultrasound performed by experienced personnel every 6 months [159,160]. Surveillance is indicated for all patients with cirrhosis. For patients with less advanced liver diseases, risk stratification using regression analysis is used to determine surveillance interval.

Models predicting the need for HCC surveillance in HCV patients use factors such as age, alcohol intake, platelet count, gamma-glutamyltransferase, and non-sustained virological response [161,162]. The ADRESS-HCC study, performed in 34,932 patients with decompensated cirrhosis from the US national liver transplant waiting list, identified six predictors of HCC (age, diabetes, race, etiology, sex, and severity of disease according to the Child–Turcotte–Pugh score) [163]. A retrospective analysis of the HALT-C trial demonstrated that the addition of the EGF SNP to clinical parameters (age, gender, smoking status, ALK-p level, and platelet count) could improve HCC risk stratification [164]. Furthermore, Ioannou et al. developed a risk stratification model for both NAFLD and ALD cirrhosis using seven predictors (age, gender, diabetes, BMI, platelet count, serum albumin, and AST/ALT ratio) [165].

HCC surveillance is often underutilized. Real-life worldwide retrospective cohorts reported screening adherence ranging from 5.7% to 78.8%, with higher rates occurring in countries with national screening programs [166]. A study from the US including 13,002 patients showed that only 42% of patients with HCV-cirrhosis received one or two surveillance tests during the first year and only 12% of them received surveillance two to four years after the diagnosis of cirrhosis [167].

Although ultrasound surveillance is currently the gold standard for HCC surveillance, there are downsides including sensitivities ranging from 47% to 84% depending on the operator’s experience [168]. Magnetic resonance imaging (MRI) has high sensitivity and specificity for diagnosis of HCC and has the potential to improve HCC surveillance outcomes. In high-risk patients with cirrhosis, surveillance by MRI using liver-specific contrast increased early HCC detection compared to ultrasound but survival benefits and cost-effectiveness have not been demonstrated [169].

## 7. Etiology-Specific Secondary Chemoprevention

### 7.1. Hepatitis B

Antiviral treatments for HBV consist mainly of interferon therapies and nucleoside/nucleotide analogs [170]. Interferon alpha (IFN-α) therapy has shown inconsistent effects on HCC prevention due to its moderate effects on HBV viral replication [171]. The beneficial effects of nucleoside/nucleotide analogs are well established. In a randomized control trial, Liaw et al. demonstrated that continuous treatment with lamivudine significantly reduced the incidence of hepatic decompensation and the risk of HCC (3.9 percent vs. 7.4 percent) in patients with advanced liver disease [172]. In a retrospective study of 872 patients versus 699 historical controls, the annual incidence of HCC was reduced from 4.1% to 0.95% in patients with sustained response to lamivudine [173]. In one systemic review of 21 studies, the incidence of HCC was significantly lowered in HBV positive patients treated with lamivudine (2.8% vs. 6.4%; *p* < 0.01) [174].

### 7.2. Hepatitis C

Direct-acting anti-virals (DAAs) targeting viral protease, polymerase, and non-structural proteins have enabled improved sustained viral response compared to interferon-based therapies [175]. DAAs are better tolerated in cirrhotic patients in comparison to interferon-based therapies [176]. HCV-related cirrhosis mortality reached a plateau in 2014 and markedly declined from 2014 to 2016 after the introduction of DAAs [177]. After treatment, the sustained virologic response (SVR) is the best indication for successful HCV treatment [178]. Janjua et al. demonstrated that among DAA-treated patients, the HCC incidence rate was 6.9% in the SVR group vs. 38.2% in the non-SVR group [179]. Ioannou et al. found that DAA-induced SVR was associated with a 71% HCC risk reduction [180]. However, patients with pre-SVR fibrosis scores ≥ 3.25 have a higher annual incidence of HCC (3.66%/year) than those with <3.25 (1.16%/year) [181]. The persistence of HCC risk after HCV treatment can be partially explained by HCV-induced epigenetic modifications [182].

HCC surveillance should be continued in high-risk patients after DAA therapy. Despite the ongoing developments in HCV treatment options, the increasing rate of infection in young adults (age < 30) and the lack of screening are significant obstacles [183]. Barriers to HCV screening include lack of awareness, mental illness, lack of access to health care, and substance misuse [184]. The US Preventive Services Task Force recommends screening for adults at high risk, as well as one-time hepatitis C screening to all individuals born between 1945 and 1965. In high risk communities, the use of non-invasive fibroscanning can potentially identify individuals with chronic HCV [185].

Lifestyle intervention can be effective in patients with NAFLD and NASH. Weight loss > 10% has been shown to induce complete regression of NASH and partial regression of fibrosis [186]. Obese, sedentary individuals have increased risks of NAFLD in comparison to weight-matched physically active individuals [187]. EASL guidelines recommend moderate-intense aerobic physical activities in 3–5 sessions for a total of 150 min per week [188]. AASLD guidelines suggest the beneficial effects of physical activity but does not specify the exercise regimen [189]. There are currently no longitudinal studies demonstrating the effects of exercise on HCC risk reduction. Various pre-clinical rodent studies have demonstrated the efficacy of exercise in delaying HCC. In a hepatocyte-specific PTEN-deficient mouse model that developed steatohepatitis and spontaneous HCC, animals randomized to exercising developed fewer HCC nodules compared to sedentary animals (71% vs. 100%, respectively) [190].

Bariatric surgery is an effective option for weight loss in patients who are refractory to conservative treatment options. It has also been shown that bariatric surgery is a potential therapy for NASH. Lassailly et al. demonstrated that NASH resolved in 85% of patients after bariatric surgery while fibrosis was reduced in 33.8% of patients [191]. A meta-analysis demonstrated that bariatric surgery was associated with improvements in steatosis (91.6%), NASH (81.3%), as well as fibrosis (65.5%) [192]. Kwak et al. found that bariatric surgery was associated with a lower risk of HCC among matched cohorts of morbidly obese patients [193].

Diabetes mellitus (DM) has been shown to be an independent risk factor for HCC development [194]. Hyperinsulinemia can stimulate liver cell proliferation via the upregulation of IGF-1 [195] as well as hepatic stellate cell activation [196]. Insulin resistance is also an independent risk factor for liver fibrosis [197]. Given that NAFLD/NASH and DM commonly exist together, it is a reasonable hypothesis that anti-diabetic drugs have potential chemopreventive effects against NAFLD/NASH induced HCC. Metformin has been shown in several non-randomized studies to have HCC preventative effects in type-2 diabetic males [198]. In pre-clinical studies, the anti-carcinogenic effects of metformin have been shown to be mediated through the upregulation of AMPK, and the subsequent inactivation of mTOR via the upstream regulator of AMPK, LKB1 [199,200]. When exposed to DEN, male rats treated with metformin developed less fibrosis, cirrhosis, and overall fewer tumor nodules [201]. Metformin has been shown to improve liver histology and ALT levels in 30% of patients with NASH (NCT00063232). However, there are no completed clinical trials to date examining the effects of metformin administration on HCC prevention. The only trial to date (NCT02319200) was terminated early due to the lack of participants.

Long-term pioglitazone treatment can improve hepatic triglyceride content and fibrosis in patients with diabetes and NASH [202]. In a standard model, mice receiving a single injection of DEN, followed by the administration of a choline deficient L-amino acid diet, developed hepatic fibrosis and HCC nodules. In this model, pioglitazone administration at the initial onset of fibrosis resulted in a reduction in fibrosis and tumor nodules [203]. Pioglitazone targets downstream nuclear hormone PPAR*γ* by binding to retinoid X receptor and subsequently regulating insulin sensitivity, glucose metabolism, and hepatic inflammation [204]. However, there are non-negligible side effects associated with pioglitazione, such as heart failure, weight gain, and bone loss.

Vitamin E is a lipid-soluble nutrient that acts as an antioxidant to prevent free radical damage in membranes and plasma lipoproteins [205]. Treatment with vitamin E has been shown to improve liver functions and fibrosis [206,207]. However, the effects of vitamin E on inflammation are controversial. A number of studies demonstrated no improvement in inflammation, while other studies concluded that vitamin E was associated with a significant improvement in steatosis, fibrosis, and inflammation [208,209,210]. No clinical data is available on vitamin E’s HCC prevention effects.

### 7.3. Alcohol

Alcohol abstinence remains to be the most important treatment for alcohol-related hepatic disease. Alcohol cessation has been shown to decrease the risk of HCC by 6–10% per year. After two decades, the risk becomes equal to the general population [211]. Among former versus current drinkers, the odds-ratio of men developing HCC was significantly higher in those who stopped drinking for less than 10 years [212].

However, targeting sobriety is both complex and difficult to maintain. The current gold standard for alcohol use disorder is achieving total abstinence and preventing relapse [213]. Both inpatient and outpatient rehabilitation programs have shown efficacy in helping patients maintain abstinence [214]. It has also been shown that participation and communication with an alcohol addiction specialist in Alcoholic Anonymous can help to maintain abstinence [215]. Cognitive-behavioral coping skills therapy (CBT) is a psychotherapeutic approach that helps patients recognize risks for relapse and develop strategies to mitigate the risks. Patients are also encouraged to keep a diary to document the risk events [216].

Disulfiram, the most common and oldest pharmaceutical intervention for alcohol use disorder, works by inhibiting aldehyde dehydrogenase, resulting in an accumulation of aldehyde that usually results in a disulfiram–alcohol reaction, consisting primarily of tachycardia, flushing, nausea, and vomiting [217]. Studies have shown that disulfiram is effective in promoting short-term abstinence [218]. Naltrexone is an agent that blocks opioid receptors, which in turn leads to a reduction in dopamine levels and a reduction in alcohol intake [219]. The Combined Pharmacotherapies and Behavioral Interventions (COMBINE) study (NCT00006206) demonstrated that naltrexone, when given with medical counseling, resulted in an increase in the days of abstinence [220].

## 8. Etiology-Independent Secondary Chemoprevention Strategies

### 8.1. Statins

Statins, 3-hydroxy-3-methylgutaryl coenzyme A reductase inhibitors, are cholesterol-lowering agents that have cardiovascular protective effects [221]. Several randomized-controlled trials have demonstrated that statins have preventative effects in colorectal [222], breast [223], and prostate cancer [224]. Atorvastatin (10 mg/day) use in biopsy proven NASH patients demonstrated a 74% improvement in liver function tests as well as a rise in serum protein and adiponectin, a key regulator of lipid metabolism [225,226]. Statin use has been shown to correlate with a decreased risk of HCC carcinogenesis and recurrence after resection [227,228,229].

The anti-neoplastic effects of statins have been attributed to the inhibition of MYC [230], AKT [231,232], NF-κB, and IL6 production [233]. Statin use also reduces hepatic stellate cell activation via the induction of sterol regulatory element-binding protein 1 and PPAR [234], as well as reduction in portal hypertension via non-canonical hedgehog signaling [235]. Secondary prevention effects of simvastatin in patients with cirrhosis are being tested in a phase II clinical trial (NCT02968810). Currently, a multi-center double-blinded randomized clinical trial of tertiary prevention is being conducted with atorvastatin vs. placebo for HCC recurrence after completion ablation or hepatic resection (SHOT trial; NCT03024684).

### 8.2. Aspirin, COX2 Inhibitors and Anti-Platelet Agents

The major risk factor for HCC carcinogenesis is the non-resolving inflammation resulting in dysregulated production of cytokines, chemokines, growth factors, prostaglandins, and ROSs [236]. It is well established that TNF-α activated NF-κB is a critical mediator for HCC carcinogenesis [237]. In a large prospective study, the use of nonsteroidal anti-inflammatory drugs (NSAIDs) among men and women between the ages of 50 and 71 years was associated with a 37% reduced risk of HCC as well as a 51% reduced risk of mortality from chronic liver disease [238]. Cyclooxygenease-2 (COX2) controlled prostaglandins are upregulated in chronic liver disease [239]. Leng et al. demonstrated that COX2 overexpression in vitro resulted in cell growth and overexpression of AKT, while treatment with COX-2 inhibitor, celecoxib, resulted in a significant reduction in AKT activation and upregulation of apoptosis [240].

In two prospective cohorts of U.S. men and women, regular use of aspirin was associated with a significant reduction in the risk of developing HCC compared to non-regular use (2.1 vs. 5.2 cases per 100,000 person-years). However, prevention effects were not observed with other NSAIDs [241]. Aspirin has also been shown to inhibit platelet thromboxane, subsequently leading to the inhibition of spingosine-1-phosphate S1P, a lipid molecule that has been shown to promote HCC proliferation [242]. A study demonstrated that the combination of aspirin and clopidogrel reduced intrahepatic immune cell infiltration, NASH, and HCC [243]. However, increased risk of bleeding may limit the use of these drugs for long-term prevention, particularly in cirrhotic patients. Besides its anti-inflammatory properties, aspirin has also been shown to have anti-fibrotic properties. Wang et al. showed that aspirin targets P4HA2, an enzyme involved in collagen synthesis [244]. Aspirin administration to mice that were subcutaneously engrafted with HepG2 cells resulted in a reduction in collagen deposition and tumor growth [245]. Daily aspirin use was also shown to significantly lower the odds-ratio of NASH and fibrosis in 361 adults with biopsy-proven NAFLD [246].

### 8.3. Anti-Fibrosis Therapy

Fibrosis has been shown to be a key risk factor for HCC [247]. However, anti-fibrotic therapies for HCC prevention have not been established. Most clinical trials are designed to study the anti-fibrosis or anti-cancer effects of drugs, but rarely both. Though promising, therapies such as ASK-1 inhibitor, selonsertib (NCT02466516) and dual PPARα/δ agonist, elafibranor (NCT02704403), have demonstrated efficacy in reducing fibrosis but have not been tested for HCC prevention [248,249].

### 8.4. Nutritional Agents

Food-derived agents, nutritional supplements, and certain phytochemicals, plant-derived bioactive chemicals, have been recognized as potential prevention options for HCC. Glycyrrhizin, an extract of licorice root, has been shown to lower serum aminotransferases, improve liver histology, and delay HCC carcinogenesis in humans and animal models [250,251,252,253]. Sho-saiko-to, a Chinese herbal medicine that contains glycyrrhizin, was shown to increase survival in cirrhotic HBV patients as well as decrease the incidence of HCC [254]. Beta-carotene derived from fruits and vegetables reduced the number and size of hepatic nodules in rats injured by DEN and phenobarbital [255]. Epigallocatechin gallate (EGCG), the most abundant green tea catechin polyphenol, has been shown to inhibit tumor growth and induce apoptosis in vitro [256]. In a rodent HCC model of DEN and aflatoxin, EGCG treatment reduced the number of placental glutathione S-transferase positive pre-neoplastic nodules [257]. In a phase 2 clinical trial, consumption of green tea polyphenols led to a significant reduction in oxidative DNA damage in HBV positive patients exposed to aflatoxin [258]. EGCG was also shown to reverse the poor prognostic gene signature described by Hoshida et al. [137,259]. The mechanisms of HCC risk reduction with coffee consumption have yet to be determined. However, coffee has been shown to contain numerous anti-carcinogenic chemical compounds. Diterpenes have been shown to upregulate detoxifying enzymes and reduce the formation of aflatoxin–DNA adducts [260].

Higher vitamin D, 25(OH)D, levels have been associated with a reduced risk of HCC, while low levels are associated with increased risk of HBV-related HCC [261,262]. In DEN-injured mice, vitamin D3 up-regulated protein 1 (VDUP1) has been shown to suppress TNF and NF-κB activation [263]. Oral vitamin D3 is currently under investigation for the prevention of HCC in HBV patients (NCT02779465). Branched-chain amino acids (BCAA) have been shown to reduce hepatic fibrosis and HCC carcinogenesis in DEN-injured rats [264]. In an observational study of cirrhotic patients in Japan, BCAA supplementation was associated with a lower incidence of HCC development [265].

Fish is a rich source for n-3 polyunsaturated fatty acids and has been shown to reduce the risk of HCC by 35% [266] irrespective of the viral hepatitis status [267]. In a NAFLD HCC rodent model, mice fed with an n-3 polyunsaturated fatty acid supplemented diet have significant reductions in fibrosis and tumor nodules [268]. However, processed red meat has been shown to actually increase the risk of HCC [269] (Figure 5).

## 9. Challenges and Obstacles in Prevention

A major obstacle in HCC chemoprevention is the lack of accurate pre-clinical models that closely mimic HCC carcinogenesis in humans. Many drugs that enter phase I clinical trials are able to progress to phase II [270]. However, 95% of drugs that enter clinical trials do not enter the market [271]. Many drugs are initially tested in the pre-clinical setting using in vitro systems. Cancer cell lines are invaluable in vitro models that are widely used for cancer research and novel drug discovery. The major concern is that they do not accurately reflect their tissues of origin due to genetic mutations and passage cycle-derived transcriptomic alterations. Many human HCC cell lines are strikingly different to their tumors of origin [272]. This is likely why compounds that show promise in in vitro models are ineffective clinically [273]. There lacks an in vitro system in cancer prevention research that captures cancer initiation, promotion, and progression. Most commonly established human HCC cell lines such as HepG3 and HuH7 can be used to investigate cancer treatment, but not prevention.

An ideal in vivo animal model should capture the key biological features of HCC and recapitulate the tumor microenvironment. Major etiologies for failure include the complex molecular heterogeneity as well as the limited understanding of HCC carcinogenesis. HCC genetically engineered mouse models (GEEMs) activate oncogenes such as *HRAS* or *MYC* [274] or disrupt tumor suppressor genes such as *PTEN* or *TP53* [275]. However, most GEMMs have failed in addressing the complex interaction between HCC and the representative microenvironment.

Chemical carcinogens such as carbon tetrachloride [276], DEN [277] and thioacetamide [278] induce fibrosis, cirrhosis and HCC sequentially. The repeated, low dose DEN cirrhosis-driven rat model [279] demonstrated an induction of the HCC prognostic gene signature similar to that of the human signature [139]. Chemical carcinogens such as DEN are an excellent model for chemoprevention research given the accurate recapitulation of all steps in the HCC carcinogenesis pathway.

Prior to the omics era, there were limited HCC prevention targets. Transcriptomic analysis of clinical specimens and reverse-engineering prevention targets may ultimately overcome this challenge. Cirrhotic patients expressing the poor prognostic signature would benefit the most from chemoprevention. HCC chemoprevention clinical trials are difficult to conduct and expensive given the requirement for a large sample size and long observation periods. Even if an agent demonstrates efficacy with a low toxicity profile, it still takes 5–10 years for a drug to move through phase III clinical trial [280]. Prevention trials focused on lifestyle modification, such as weight control, diet, and physical activity, can be challenging because these activities are typically clustered, thus identifying the change of a single behavior can be difficult if not impossible [281].

Another major challenge in HCC prevention is the lack of understanding of the long term tolerability and impact on quality of life of many drugs. To be acceptable for HCC prevention, a drug needs to be well tolerated for an extended period of time with minimal—if any—side effects. It is also important for physicians to be comfortable in prescribing these medications. Even minor side effects can be enough to affect quality of life and compliance with a cancer prevention medication, thus limiting its efficacy. This is especially of importance in HCC prevention, where a majority of this patient population have some form of chronic cirrhosis. Metformin is generally well tolerated with a good safety profile. However, lactic acidosis, one of its most notorious side effects, is more likely to occur in patients with hepatic insufficiency [282]. Aspirin is also commonly prescribed and well tolerated, but physicians may not be comfortable prescribing aspirin to cirrhotic patients who are at high risk of gastrointestinal bleeds.

## 10. Conclusions

Integration of HCC prevention research to the clinical setting is an extremely important strategy. Prevention clinical trials are very challenging to conduct because of the need for large sample sizes and long observation times. In addition, establishing the optimal dose and duration for chemopreventive drugs remains a challenge. Although there has been notable success in primary and secondary prevention for viral hepatitis such as the HBV vaccine or HCV cure by DAAs, similar successes for prevention of metabolic HCC are largely absent given the challenges associated with strategies of improved diet and regular exercise. Furthermore, there is no approved strategy to prevent HCC in advanced fibrosis or post HCC resection. Advances in the field of HCC chemoprevention will be aided by a more complete characterization of HCC carcinogenesis as well as a better understanding of the liver microenvironment. The main challenge in HCC prevention research will always be translating pre-clinical research into successful clinical trials but there is a promise for success as we develop more individualized therapies.

## Figures and Tables

**Figure 1 jcm-09-03817-f001:**
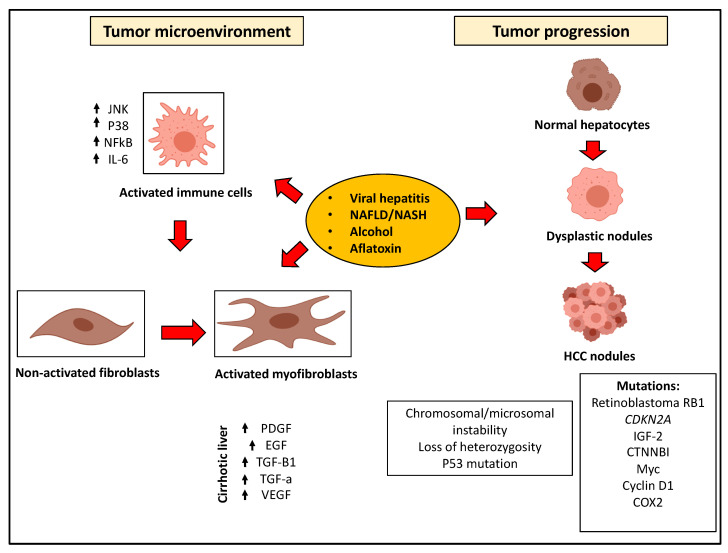
Mechanisms of hepatocellular carcinoma. Molecular pathways of HCC carcinogenesis are summarized. Risk factors include viral hepatitis, NAFLD, alcohol and toxins. HCC tumors develop as dysplastic nodules through the gaining of molecular aberrations and mutations. The cirrhotic microenvironment in the liver promotes HCC carcinogenesis through the activation of hepatic stellate cells into myofibroblasts. The cirrhotic background also promotes inflammation leading to the upregulation of pro-carcinogenic genes and pathways (text for details).

**Figure 2 jcm-09-03817-f002:**
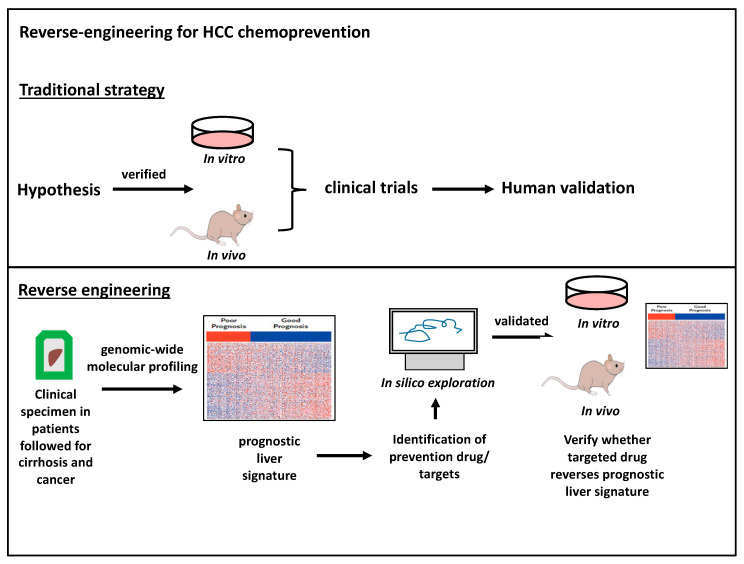
Reverse engineering for HCC chemoprevention. Traditionally, chemoprevention targets are verified in both in vitro and experimental animal models and then introduced into clinical trials (top panel). The reverse-engineering identifies targets for chemoprevention in human cohorts already followed for decades. Samples are genetically profiled into molecular signatures and then experimentally evaluated for mechanisms and therapeutic strategies (bottom panel).

**Figure 3 jcm-09-03817-f003:**
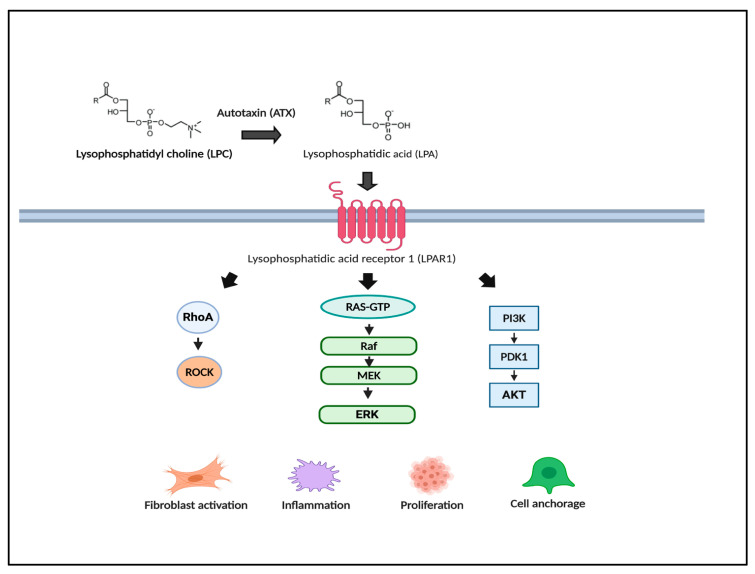
Lysophosphatidic acid (LPA) pathway as a novel chemoprevention target. Reverse engineering transcriptome analysis revealed the LPA pathway as a target for HCC prevention. LPA and its G protein-coupled receptors, lysophosphatidic acid receptors (LPARs), promote fibrosis, inflammation, and carcinogenesis through the activation of down-stream RhoA, Ras/mitogen-activated protein kinase (MAPK)/extracellular-signal-regulated kinase (ERK), and Akt/PI3K. LPA activation has been observed in human and rodent cirrhotic livers at risk for HCC.

**Figure 4 jcm-09-03817-f004:**
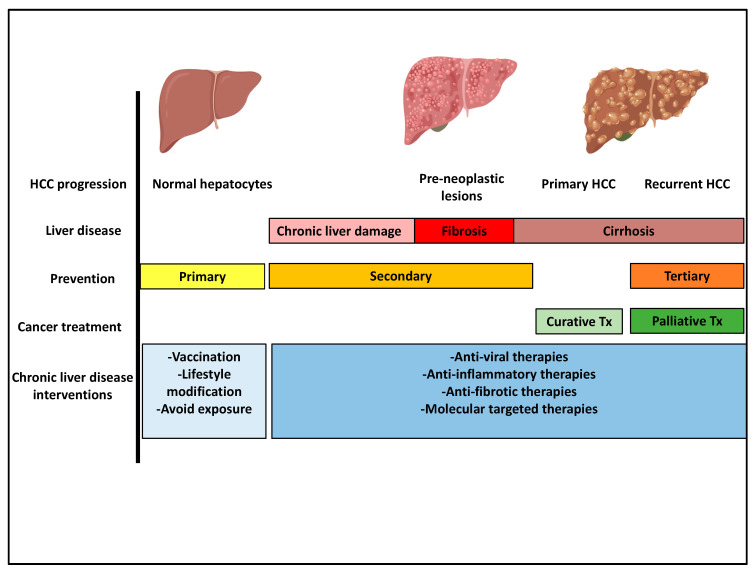
HCC-prevention strategies through the progression of HCC development. HCC prevention strategies, primary, secondary, and tertiary prevention, target various stages of liver disease progression (text for details).

**Figure 5 jcm-09-03817-f005:**
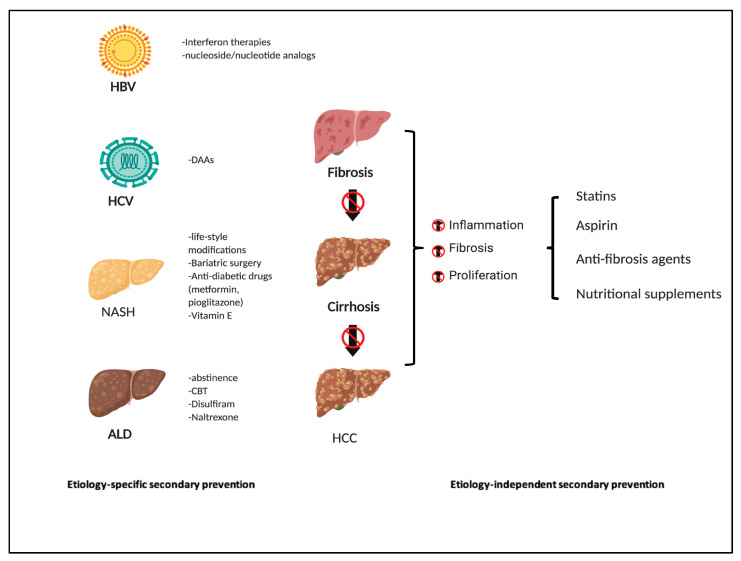
Etiology-specific and independent secondary HCC prevention. Etiology-specific secondary prevention targets the various risk factors of HCC development, including HBV, HCV, NAFLD/NASH, and ALD. Etiology independent prevention strategies include agents that have anti-fibrosis or anti-inflammatory activities.

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
