# Peer review of "Risk Factors, Pathogenesis, and Strategies for Hepatocellular Carcinoma Prevention: Emphasis on Secondary Prevention and Its Translational Challenges"

_jcm, 2020, doi:10.3390/jcm9123817_

Round 1

Reviewer 1 Report

The review of Li et al. have done an excellent job of presenting the current state of hepatocellular carcinoma (HCC) in terms of disease, disease progression, causes, treatments and outcomes. This review represents a complete survey of all the relevant issues and does so in a manner that is very pleasing to read. I have only minor comments.  
  1. One major omission is the involvement of lipid droplets (LDs). This important issue could be incorporated in multiple sections. LDs are necessary for viral replication. NS5A binds the major Rab protein (Rab18) of LDs. Impairment of lipophagy of LDs leads from NAFLD to NASH. Accumulation of sugar derived FAs/TGs in LDs is the driving cause of NAFLD and a complicating factor in viral mediated liver damage.  These are the types of comments that could be added.
  2.  Very nice figures, very clear and nice use of colour.
  3. Line 425: State target of pioglitazone
  4. Line 432: nutrient

Author Response

1. One major omission is the involvement of lipid droplets (LDs). This important issue could be incorporated in multiple sections. LDs are necessary for viral replication. NS5A binds the major Rab protein (Rab18) of LDs. Impairment of lipophagy of LDs leads from NAFLD to NASH. Accumulation of sugar derived FAs/TGs in LDs is the driving cause of NAFLD and a complicating factor in viral mediated liver damage.  These are the types of comments that could be added.

Comment on lipid droplets (LDs) greatly appreciated. This was incorporated into the viral hepatitis section Thank you for the comments.

2. Very nice figures, very clear and nice use of color.

Thank you for the comment.

3. Line 425: State target of pioglitazone

Thank you for the comment. We have discussed the downstream target of pioglitazone, PPAR-Y.

4. Line 432: nutrient

Thank you for the comment.

Reviewer 2 Report

GENERAL COMMENT

This review article addresses risk factors, pathogenesis, and strategies of HCC prevention focusing on secondary prevention and translational challenges. While its points of strength include the topic being a major clinical challange, this submission has some problems in consistency in writing (e.g. the tile). Introduction needs reworking. Moreover, as a typically sexually dimorphic disease, HCC needs to be described as such (Lancet. 2020 Aug 22;396(10250):565-582) and additional epidemiological and biological data need to be added regarding this key feature of disease. Among the various risk factors for NAFLD and NASH development, hypothyroidism has he potential to progress to HCC and this must be discussed. Similarly, specific aspects of lifestyle such as diet and physical activity should be discussed. Early diagnosis is a key factor affecting the chances for a radical management of HCC and the authors may be willing to address this. Some paradigms appear to be somewhat outdated owing to either incomplete or old references. Finally, the section on statins and glucose-lowering drugs must be expanded.

SPECIFIC COMMENT

MAJOR

Title – As presently written, it seems to suggest that the focus is on the pathogenesis (and clinical challenges) of secondary prevention which may hardly be the case. Therefore, to avoid any misunderstandings, the title should possibly read: Risk factors, pathogenesis, and strategies of HCC prevention: focus on secondary prevention and translational challenges.

Introduction – too long and poorly focused on the specific aim(s). Please give compelling reasons supporting the choice of this specific topic. Lines 43-50 should strongly be sumarized by deleting any allusion to specific mechanisms of carcinogenesis given that these are covered elsewhere throughout the text.

HCC as a sexually dimorphic disease – These authors declare that HCC is over-represented among men as compared to women. Add a paragraph on the biological bases of this sexual dimorphism notably including the role of IL-6 and sex hormones. This is pathogenically relevant and may possibly be a clue to discovering novel preventive/treatment strategies.

Endocrine risk factors – A specific line of research has described NAFLD/NASH secondary to endocrine disorders (Int J Mol Sci. 2019 Jun 11;20(11):2841). Among these, hypothyroidism has been reported to have the potential for progressing to HCC (Dig Liver Dis. 2019 Apr;51(4):462-470). This topic needs to be addressed in full molecular detail (Sci Rep. 2017 Nov 1;7(1):14868. J Hepatol. 2020 Jun;72(6):1159-1169) given that it may disclose specific preventive strategies.

Lifestyle The first citation of this article states "Coffee reduces risk for hepatocellular carcinoma" However, this is not even mentioned in the text. Similarly, studies have shown that a diet based on fish may reduce HCC risk and this is not even mentioned (Cancer Causes Control. 2015 Mar;26(3):367-76 Int J Epidemiol. 2019 Dec 1;48(6):1863-1871). Along the same line the role, if any, of meat consumption should be discussed (Int J Epidemiol. 2019 Dec 1;48(6):1863-1871). And what about the role of smoking and obesity ? (Eur J Cancer Prev. 2018 May;27(3):205-212. Ann Hepatol. 2019 Nov-Dec;18(6):810-815). The role of alcohol consumption in those with NAFLD must also be discussed (J Gastroenterol Hepatol. 2020 May;35(5):862-869).

Early diagnosis is a key factor affecting the chances for a radical management of HCC – A paragraph devoted to points of strenghts and limitations of this approach should be included. Reference should be made to novel models available for risk stratification based on the etiology of liver disease (J Hepatol. 2019;71:523-533). Please, address strengths and weaknesses of conventional ultrasonography, which remains a widely available and affordable tool in many areas worldwide.

References – tend to be incomplete and outdated. For example References 146 and 147 must be deleted and this section totally reworked based on the following: Hepatology. 2020;71:1910-1922. J Viral Hepat. 2020 Aug;27(8):781-793. Curr Treat Options Gastroenterol. 2018 Jun;16(2):203-214. J Hepatol. 2017 Sep 5:S0168-8278(17)32273-0. J Hepatol. 2017 Dec;67(6):1204-1212. In addition, a seminal paper should also be cited and discussed Gut. 2016 May;65(5):861-9.

Drugs of potential benefit in preventing HCC – the section on statins and glucose-lowering drugs must be expanded as to include the activity, if any, of these on NASH histology, an acknowledged precursor lesion/risk factor for the development of HCC (Metabolism. 2017 Jun;71:17-32. Atherosclerosis. 2019 May;284:66-74. Acta Diabetol. 2019 Apr;56(4):385-396).

MINOR

Throughout the manuscript rename "non-alcoholic" to "nonalcoholic" in agreement with History of NAFLD.

Round 2

Reviewer 2 Report

I am pleased in reporting that this submission is improved as a result of these Authors taking into consideration all those suggestions that were aimed at improving their submission.

As a minor change I would like to report that Lines 397-400  in the text need attention "Crowley D, Cullen W, Laird E, Lambert JS, Mc Hugh T, Murphy C, et al. Exploring patient characteristics and barriers to hepatitis C treatment in patients on opioid substitution treatment  attending a community based fibro-scanning clinic. J Transl Intern Med. 2017;5:112Nonalcoholic fatty liver disease (NAFLD)/nonalcoholic steatohepatitis (NASH)"
